# Introduction of Nonacidic Side Chains on 6-Ethylcholane Scaffolds in the Identification of Potent Bile Acid Receptor Agonists with Improved Pharmacokinetic Properties

**DOI:** 10.3390/molecules24061043

**Published:** 2019-03-16

**Authors:** Claudia Finamore, Giuliana Baronissi, Silvia Marchianò, Francesco Saverio Di Leva, Adriana Carino, Maria Chiara Monti, Vittorio Limongelli, Angela Zampella, Stefano Fiorucci, Valentina Sepe

**Affiliations:** 1Department of Pharmacy, University of Naples "Federico II", via D. Montesano 49, 80131 Naples, Italy; claudia.finamore@unina.it (C.F.); giuliana.baronissi@unina.it (G.B.); francesco.dileva@unina.it (F.S.D.L.); vittoriolimongelli@gmail.com (V.L.); azampell@unina.it (A.Z.); 2Department of Surgery and Biomedical Sciences, Nuova Facoltà di Medicina, Piazza Lucio Severi, 1 - 06132 Perugia, Italy; silvia4as@hotmail.it (S.M.); adriana.carino@hotmail.it (A.C.); stefano.fiorucci@unipg.it (S.F.); 3Department of Pharmacy, University of Salerno, Via Giovanni Paolo II, 132, 84084 Fisciano, Salerno, Italy; mcmonti@unisa.it; 4Università della Svizzera italiana (USI), Faculty of Biomedical Sciences, Institute of Computational Science - Center for Computational Medicine in Cardiology, Via G. Buffi 13, CH-6900 Lugano, Switzerland

**Keywords:** FXR agonists, bile acid receptors, steroidal scaffolds, medicinal chemistry

## Abstract

As a cellular bile acid sensor, farnesoid X receptor (FXR) and the membrane G-coupled receptor (GPBAR1) participate in maintaining bile acid, lipid, and glucose homeostasis. To date, several selective and dual agonists have been developed as promising pharmacological approach to metabolic disorders, with most of them possessing an acidic conjugable function that might compromise their pharmacokinetic distribution. Here, guided by docking calculations, nonacidic 6-ethyl cholane derivatives have been prepared. In vitro pharmacological characterization resulted in the identification of bile acid receptor modulators with improved pharmacokinetic properties.

## 1. Introduction

Farnesoid X receptor (FXR) belongs to the super-family of ligand-activated nuclear transcription factor proteins. Mostly expressed in enterohepatic tissues, FXR functions as bile acid [1,2,3,4] sensor, with chenodeoxycholic acid (CDCA) (Figure 1) identified as the most potent ligand in human, tuning their intracellular concentration through changes in gene expression (EC_50_ = 20 µM) [5]. In addition, FXR also regulates glucose and lipid homeostasis [6], and exerts strong anti-inflammatory activity in intestine and in the liver [7].

Thus, FXR is today recognized as a potential target for novel pharmacotherapies addressing bile acid, lipid, and carbohydrate dysregulation, such as cholestasis, liver fibrosis, steatohepatitis, diabetes (T2DM), as well as obesity, metabolic syndrome, and inflammatory bowel disease [6,8,9].

In addition, bile acids signal through the membrane G-coupled receptor GPBAR1 (TGR5/MBAR) [10], with taurolithocholic acid (TLCA) (Figure 1), the most potent endogenous activator (EC_50_ = 0.33 µM) [11]. GPBAR1 activation affects energy expenditure [12], glucose metabolism, and insulin sensitivity [13]; thus, the exogenous modulation of the receptor representing an attractive strategy to treat metabolic disorders [6,14,15,16].

Therefore, in the window of metabolic disorders, the development of ligands covering steroidal and non-steroidal chemical space endowed with dual activity toward GPBAR1 and FXR appears to be a promising strategy in non-alcoholic steatohepatitis (NASH), hypercholesterolemia, hypertriglyceridemia, and T2DM [17,18,19,20].

In the frame of steroidal family, the introduction of an ethyl group at C-6 on the CDCA ring B afforded the disclosure of 6-ethylchenodeoxycholic acid (6-ECDCA/INT-747/obeticholic acid/Ocaliva/OCA, Figure 1) [21], the most potent steroidal FXR agonist generated so far, that showed also a GPBAR1 activity (EC_50_ = 0.5 µM and 0.9 µM, respectively) [16]. OCA represents the first-in-class of the FXR ligands approved for the treatment of ursodeoxycholic acid (UDCA)-resistant patients with primary biliary cholangitis (PBC), and has progressed in Phase III trials on NASH patients (FLINT study, and REGENERATE) [22]. However, adverse drug effects emerged from safety data in FLINT trial, such as increased low-density lipoprotein cholesterol (LDL-C) and total cholesterol, decreased high-density lipoprotein cholesterol (HDL-C), and pruritus that collectively might limit OCA market application. In addition, the carboxyl group on the side chain conjugates with taurine in mice and glycine in humans, and the corresponding conjugated metabolite undergoes to extensive enterohepatic circulation, resulting in the risk of drug accumulation. In this context, with the aim to avoid glyco- or tauro-conjugation at carboxylic end group, promising results have been obtained preserving the 6α-ethyl group and the 7α-hydroxyl group on the ring B of CDCA, and introducing non-conjugable functional groups on the side chain such as the C-23 alcoholic function in norECDCOH (Figure 1) [16], or carboxylic acid bioisosteres (amide, urea, and sulfonamide derivatives) [22].

Here, compounds **1**–**6** (Table 1) were designed to introduce a nonacidic side chain on 6-ethyl cholane scaffold, thus expanding the chemical diversity of available bile acid receptor modulators and identifying efficacious agonists with improved pharmacokinetic properties.

## 2. Results and Discussion

### 2.1. Chemical Synthesis and Evaluation of Biological Activity

Synthesis started with the key intermediate 3α-hydroxy-6α-ethyl-7-keto-5β-cholan-24-oic acid **7**, prepared in a very efficient multi-step procedure starting from 7-ketolithocholic acid (7-KLCA, 67% over six steps) [23]. Acetylation at C-3, oxidative decarboxylation [24], followed by deacetylation furnished compound **8** (58% over three steps). Reduction at C-7 gave **1** (98%), that was in turn transformed in quantitative yields in **2** and **3** through double bond hydrogenation and reductive ozonolysis, respectively (Scheme 1).

The newly synthesized analogues were evaluated on human orthologs of FXR and GPBAR1 receptors in a luciferase reporter assay. As shown in Table 1, the introduction of a nonacidic side chain on the 6-ethylchenodeoxycholic scaffold produced beneficial effects on FXR with compounds **1**, **2**, and **3** emerging as high efficacious FXR agonists in transactivation assays, with the above efficacy data translating also in good EC_50_ values in cell-assays.

Even if compounds **1**, **2**, and **3** were demonstrated less efficacious in transactivating GPBAR1 (55–79% range vs TLCA), the results in term of potency could be judged promising with **1** and **3**, two high potent GPBAR1 agonists. Collectively, compounds **1**, **2**, and **3** emerged encouraging dual agonists.

Having successfully identified nonacidic side chain derivatives with good effect in term of potency and favorable impact on metabolism (see below and Table 2), optimization of selectivity was in focus. FXR selectivity is a critical point in the pharmacological claim of FXR agonists in cholestasis disorders, where the concomitant activation of GPBAR1, recently demonstrated the physiological mediator of itching in mice [25], limits the pharmacological utility of dual agonists.

Thus, following our recent observation that the removal of the hydroxyl group at C-3 on 6-ECDCA scaffold is detrimental in term of GPBAR1 agonism and produces potent and selective FXR agonists (3-deoxy-6-ECDCA in Figure 1) [23,26], compounds **4**–**6** have been prepared according to Scheme 2.

Basic hydrolysis on methyl 6α-ethyl-7-keto-5β-cholan-24-oate **9** [23] furnished the corresponding carboxylic acid **10** in quantitative yield. Oxidative decarboxylation and LiBH_4_ reduction at C-7 provided the alkene **4** (91%). Double bond hydrogenation and reductive ozonolysis gave **5** and **6** in quantitative and 95% yields, respectively.

As expected, results of transactivations of cAMP response element-binding protein (CREB) in HEK293T, transiently transfected with the membrane bile acid receptor GPBAR1 (Table 1), revealed that the strategy of elimination at C-3 hydroxyl group produces inactive derivatives at 10 μM concentration. Surprisingly, the elimination of the hydroxyl group at C-3 with the concomitant introduction of a nonacidic side chain also negatively affects FXR agonism, with a general loss in efficacy and potency. Among the above subset, only compound **6** maintains a promising efficacy when tested at 10 μM, even if the above efficacy translates in a moderate potency (EC_50_ = 13.7 µM). Indeed, the lack of the OH at C-3 position as well as the modification on the side chain produces an impressive improvement in term of metabolic stability (Table 2), due to the loss of the two well-known primary points of metabolization (sulfation at C3 and tauro/glyco conjugation at C24) on bile acid scaffold as reported by Alnouti [27].

Physicochemical properties of the active compounds are summarized in Table 2. As shown, all nonacidic 6-ethyl cholane derivatives prepared in this study showed a comparable aqueous solubility to that of 6-ECDCA whereas they are endowed with improved metabolic stability as assessed by incubation with rat liver microsomes [28]. Indeed, the intrinsic clearance (Cl_int_) and half-life (t_1/2_), which represent the intrinsic ability of hepatic enzymes to metabolize the drug and its elimination half-life, respectively, show promising results for **3** (60.6% remaining after 40 min incubation, corresponding to a Cl_int_ of 36.6 µL/min/mg of microsomal proteins and t_1/2_ of 63.0 min) and its 3-deoxy counterpart **6** (79.2% remaining after 40 min incubation, corresponding to a Cl_int_ of 19.7 µL/min/mg of microsomal proteins and t_1/2_ of 117.4 min).

We have subsequently analyzed whether compounds **1**–**3** and **6** regulate canonical functions exerted by FXR and GPBAR1, by reverse transcriptase-polymerase chain reaction (RT-PCR) analysis. As shown in Figure 2A, compounds **1**–**3** were able to induce the expression of pro-glucagon mRNA in GLUTag cells, an intestinal endocrine cell line that expresses physiologically GPBAR1, while **6** failed to regulate this canonical GPBAR1 function.

In vitro characterization of small heterodimer partner (SHP) mRNA, a canonical FXR target gene was carried out by assessing the response of human HepG2 cells, expressing physiologically FXR, primed with compounds **1**–**3** and **6**. As illustrated in Figure 2B, compounds **1**–**3** and **6** were able to induce SHP mRNA in a dose dependent manner with compound **6** more potent than CDCA (10 µM) in the induction the above expression, when administered at 10 μM concentration. Collectively, these results are fully consistent with the nature of compounds **1**–**3** as dual FXR/GPBAR1 agonists and compound **6** as a potent and selective FXR agonist.

### 2.2. Molecular Docking

In order to elucidate the binding mode of our newly synthesized bile acid derivatives, we performed docking calculations that is a widely used computational technique to generate and rank ligand/protein complexes based on scoring functions [29,30,31]. In particular, we investigated through molecular docking calculations the binding mode to FXR and GPBAR1 of the dual agonist **3**. In FXR, the best scored docking pose (Figure 3A) reveals that the steroidal scaffold of **3** establishes favorable hydrophobic interactions with the side chains of residues such as Leu284, Met287, Ala288, Met325, Phe333, Leu345, and Ile349.

Moreover, the short alcoholic side chain of the ligand is found in an amphipathic region of the FXR-ligand binding domain (FXR-LBD), where it might form water-mediated hydrogen bonds with the side chains of His291, Arg328, and Ser329. In fact, the formation of polar interactions (i.e., with Arg328) at this site can contribute to FXR activation, as previously reported in literature [32]. On the other side, the ligand’s 3α-OH engages H-bonds with the side chains of both Tyr358 and His444. Remarkably, the formation of an H-bond with His444 is known to reinforce the cation-π interaction formed by this amino acid with Trp466, which stabilizes the receptor agonist conformation [32]. Finally, the ligand’s 7α-OH group engages H-bonds with Ser329 and Tyr366, while the 6α-ethyl moiety forms hydrophobic contacts with Tyr358, Ile359, and Phe363. Interestingly, the binding pose of **3** is super-imposable with that of the parent compound 6-ECDCA [32], although the latter can establish a salt bridge with Arg328 through its carboxylic side chain. Interestingly, compounds **1** and **2**, which are endowed with hydrophobic side chains and thus cannot form polar contacts with Arg328, show however a FXR agonist profile similar to **3**. In this case, the loss of the polar interaction is compensated by hydrophobic contacts formed by the ligand’s side chain with residues Met262, Met287 and Ile332 (see the Appendix A). Even compound **6** displays an efficacy comparable to that of **3**, although it lacks the 3α-OH group and hence cannot form the H-bond with His444 (see the Appendix A). This finding is in line with our recent study showing that bile acids lacking the 3α-OH group can however stabilize the cation-π interaction between His444 and Trp466 and in turn the FXR agonist conformation, through a network of hydrophobic contacts [26]. However, compounds missing both the 3α-OH and a polar side chain (**4** and **5**) can interact neither with Arg328 nor with His444, resulting inactive towards FXR. Based on our results, we can conclude that the positioning of the ligand’s ring A at the FXR binding site and the shape complementarity more than the formation of specific H-bonds, are the driving force of the ligand binding. However, an anchor point formed by either the 3α-OH group or a polar functional group on the side chain is necessary for the receptor activation.

In GPBAR1, the best scored docking pose (Figure 3B) shows that **3** binds to GPBAR1 similarly to other bile acids reported by us as agonists of this receptor [17,33,34].

In detail, the ligand’s steroidal scaffold establishes a number of hydrophobic contacts with the side chains of Leu71, Phe96, Leu174, and Trp237, while the ligand’s 3α- and 7α-hydroxyl groups engage H-bond interactions with Glu169 and Asn93, respectively. Particularly the former interaction is required for the binding and the activation of the receptor; indeed, compounds **4**–**6**, which are deprived of the 3α-OH group, are inactive towards GPBAR1. On the other side of the binding cavity, the hydroxyl side chain of **3** is oriented towards the polar cavity defined by transmembrane (TM) helices 1, 2, and 7, where it establishes a H-bond with Ser270. At variance with **3**, compounds **1** and **2** present a hydrophobic side chain that occupies the neighboring lipophilic pocket forming favorable contacts with Ala66, Leu68, and Leu71 on TM2 (see the Appendix A). Such interactions stabilize the binding of these compounds that show an efficacy comparable to **3**. This finding is in line with our previous work in which we reported compounds bearing small hydrophobic side chains with a good GPBAR1 activation profile [35].

## 3. Materials and Methods

### 3.1. General Information

High-resolution electrospray ionization mass spectrometry (ESI-MS) spectra () were performed with a Micromass Q-TOF mass spectrometer (Q-TOF premier, Waters Co., Milford, MA, USA). High-performance liquid chromatography (HPLC, Phenomenex Inc, Torrance, CA, USA) was performed using a Waters Model 510 pump equipped with Waters rheodyne injector and a differential refractometer, model 401 (Waters Co., Milford, MA, USA). Nuclear magnetic resonance (NMR) spectra were obtained on Varian Inova 400 NMR spectrometer (^1^H at 400 MHz, ^13^C at 100 MHz) equipped with a Sun hardware and recorded in CDCl_3_ (δ_H_ = 7.26 and δ_C_ = 77.0 ppm) and CD_3_OD (δ_H_ = 3.30 and δ_C_ = 49.0 ppm). *J* are in hertz (Hz) and chemical shifts (δ) are reported in ppm and referred to CHCl_3_ and CHD_2_OD as internal standards. Reaction progress was monitored via thin-layer chromatography (TLC) on Alugram^®^ silica gel G/UV254 plates. All chemicals were obtained from Zentek, S.r.l. Solvents and reagents were used as supplied from commercial sources with the following exceptions. Tetrahydrofuran, toluene, CH_2_Cl_2_, were distilled from calcium hydride immediately prior to use. Methanol was dried from magnesium methoxide as follow. Magnesium turnings (1 g) and iodine (0.1 g) were refluxed in a small (5–10 mL) quantity of methanol until all of the magnesium has reacted. The mixture was diluted (up to 100 mL) with reagent grade methanol, refluxed for 2–3 h then distilled under nitrogen. All reactions were carried out under argon atmosphere using flame-dried glassware. The purity of synthesized compounds was determined by HPLC. All compounds for biological testing were >95% pure.

### 3.2. Synthetic Procedures

#### 3.2.1. Synthesis of 6α-Ethyl-3α-hydroxy-7-keto-24-nor-5β-chol-23-ene (**8**)

A portion of 6α-ethyl-3α-hydroxy-7-keto-5β-cholan-24-oic acid **7**, obtained as previously reported (2.0 g, 4.7 mmol) [23] was acetylated with acetic anhydride (2.2 mL, 23.5 mmol) in a solution of dry pyridine (30 mL). The mixture was stirred for 5 h. After this time, the resulting solution was evaporated under reduced pressure to remove the pyridine. Then the solution was extracted with ethyl acetate (3 × 50 mL). The collected organic phases were dried over Na_2_SO_4_ anhydrous and evaporated under reduced pressure to give a residue that was subjected to the next reaction without further purification. The mixture (2.0 g, 4.34 mmol) was dissolved in dry toluene/dry pyridine (20 mL: 200 µL, 10:1 *v*/*v*) and Cu(OAc)_2_^.^H_2_O (2.6 g, 13.0 mmol) was added in dark. After 30 min, Pb(OAc)_4_ (9.6 g, 21.7 mmol) was added in dark. After 3 h the solution was heated to reflux for 1 h (no longer in the dark). The mixture was cooled, and aqueous ethylene glycol was added. The resulting mixture was extracted with diethyl ether (3 × 50 mL). The combined organic phases were washed then with saturated solution of NaHCO_3_, water and brine. After drying over anhydrous Na_2_SO_4_, the residue was evaporated under vacuum to give 1.2 g of product as crude residue. Then, the residue (1.2 g, 2.9 mmol) was treated with CH_3_ONa (156 mg, 2.9 mmol) in dry MeOH (10 mL). After stirring for 12 h at room temperature, water was added and methanol was evaporated. The residue was extracted with EtOAc (3 × 50 mL). The combined organic layers were washed with brine, dried and evaporated to dryness to give a mixture that was purified on flash chromatography, using 95:5 hexane/ethyl acetate and 0.5% of TEA, as eluent, and affording 1.25 g of pure compound **8** (58% over three steps).

Selected ^1^H-NMR (400 MHz, CDCl_3_): δ 5.65 (1H, m), 4.87 (1H, d, *J* = 17.0 Hz), 4.80 (1H, d, *J* = 10.3 Hz), 3.50 (1H, m), 1.20 (3H, s), 1.01 (3H, d, *J* = 7.3 Hz), 0.78 (3H, t, *J* = 7.4 Hz), 0.67 (3H, s); ^13^C-NMR (100 MHz, CDCl_3_): δ 212.8, 114.9, 111.6, 71.1, 54.4, 51.9, 50.6, 49.9, 49.0, 43.7, 42.5, 41.0, 38.9, 35.6, 34.2, 31.7, 29.8, 28.5, 24.6, 23.5, 21.8, 20.1, 18.8, 12.2, 11.9. HR ESIMS *m*/*z* 373.3105 [M + H]^+^, C_25_H_41_O_2_ requires 373.3101.

#### 3.2.2. Synthesis of 6α-Ethyl-3α, 7α-dihydroxy-24-nor-5β-chol-23-ene (**1**)

To a solution of **8** (1.0 g, 2.7 mmol) in dry THF (30 mL), at 0 °C dry methanol (750 µL, 18.7 mmol) and LiBH_4_ (9.3 mL, 2 M in THF, 18.7 mmol) were added. The resulting mixture was stirred for 2 h at 0 °C. The mixture was quenched by addition of 1M NaOH (5.4 mL) and then ethyl acetate. The organic phase was washed with water, dried (Na_2_SO_4_) and concentrated giving 980 mg of crude residue (98% yield). An analytic sample of compound **1** was obtained through purification with HPLC on a Luna Omega Polar C18 (5 µm; 4.6 mm i.d. × 250 mm), with MeOH/H_2_O (92:8) as eluent (flow rate 1 mL/min) (t_R_ = 14 min).

Selected ^1^H-NMR (400 MHz, CDCl_3_): δ 5.68 (1H, m), 4.91 (1H, dd, *J* = 16.8, 1.8 Hz), 4.83 (1H, dd, *J* = 10.2, 1.8 Hz), 3.71 (1H, br s), 3.41 (1H, m), 1.05 (3H, d, *J* = 6.3 Hz), 0.91 (3H, s), 0.91 (3H, t, *J* = 7.2 Hz), 0.70 (3H, s); ^13^C-NMR (100 MHz, CDCl_3_): δ 145.2, 111.6, 72.3, 70.9, 55.3, 50.5, 45.2, 42.6, 41.2, 41.1, 39.9, 39.5, 35.5 (2C), 33.9, 33.2, 30.6, 28.5, 23.7, 23.2, 22.2, 20.7, 20.1, 11.9, 11.6. HR ESIMS *m*/*z* 375.3255 [M + H]^+^, C_25_H_43_O_2_ requires 375.3258.

#### 3.2.3. Synthesis of 6α-Ethyl-3α, 7α -dihydroxy-24-nor-5β-cholane (**2**)

A solution of **1** (300 mg, 0.80 mmol) in dry THF/dry MeOH (30 mL, 1:1 *v*/*v*) was hydrogenated in presence of Pd(OH)_2_/C 20 wt % on activated carbon (100 mg) degussa type. The mixture was flushed with nitrogen and then with hydrogen several times. The reaction was stirred at room temperature over night. The catalyst was filtered through Celite, and the recovered filtrate was concentrated under vacuum to give compound as crude product (320 mg, quantitative yield). HPLC on a Luna Omega Polar C18 (5 µm; 4.6 mm i.d. × 250 mm), with MeOH/H_2_O (92:8) as eluent (flow rate 1 mL/min) furnished pure compound **2** (t_R_ = 16 min).

Selected ^1^H-NMR (400 MHz, CD_3_OD): δ 3.66 (1H, br s), 3.31 (1H, m ovl), 0.94 (3H, d, *J* = 6.5 Hz), 0.92 (3H, s), 0.90 (3H, t, *J* = 7.1 Hz), 0.85 (3H, t, *J* = 7.6 Hz), 0.70 (3H, s); ^13^C-NMR (100 MHz, CD_3_OD): δ 73.2, 71.2, 57.2, 51.6, 46.9, 43.6, 43.2, 41.5, 41.0, 38.5, 36.8, 36.6, 34.5, 34.4, 31.2, 29.4, 29.3, 24.6, 23.7, 23.5, 22.0, 18.6, 12.3, 12.0, 10.7. HR ESIMS *m*/*z* 377.3416 [M + H]^+^, C_25_H_45_O_2_ requires 377.3414.

#### 3.2.4. Synthesis of 6α-Ethyl-3α, 7α -dihydroxy-23,24-dinor-5β-cholan-22-ol (**3**)

A stream of O_3_ was bubbled into a solution of **1** (300 mg, 0.80 mmol) in dry CH_2_Cl_2_ (7 mL) kept at −78 °C until a blue-color solution resulted. After stirring for 1 min, excess of O_3_ was removed upon bubbling N_2_ and to the colorless solution was added dry MeOH (5 mL) followed by an excess of NaBH_4_. After stirring at −78 °C for 2 h, the reaction mixture was left to warm to room temperature and treated with dry MeOH (1 mL). The solution was concentrated and the resulting mixture was partitioned between EtOAc and H_2_O (3 × 50 mL). The organic phase was evaporated to give the corresponding alcohol (298 mg, quantitative yield). Purification by HPLC on a Luna Omega Polar C18 (5 µm; 4.6 mm i.d. × 250 mm), with MeOH/H_2_O (86:14) as eluent (flow rate 1 mL/min), furnished a pure aliquot of compound **3** (t_R_ = 12 min).

Selected ^1^H-NMR (400 MHz, CDCl_3_): δ 3.70 (1H, br s), 3.65 (1H, m), 3.49 (1H, br s), 3.37 (1H, m), 1.05 (3H, d, *J* = 6.7 Hz), 0.90 (3H, t, *J* = 7.1 Hz), 0.89 (3H, s), 0.68 (3H, s); ^13^C-NMR (100 MHz, CDCl_3_): δ 72.2, 70.8, 68.0, 52.5, 50.3, 45.1, 42.8, 41.2, 40.1, 39.5, 38.8, 35.5 (2C), 34.0, 33.2, 30.7, 27.8, 23.8, 23.1, 22.2, 20.7, 16.7, 11.8, 11.6. HR ESIMS *m*/*z* 379.3205 [M + H]^+^, C_24_H_43_O_3_ requires 379.3207.

#### 3.2.5. Synthesis of 6α-Ethyl-7-keto-5β-cholan-24-oic acid (**10**)

Hydrolysis with NaOH was performed on a methyl 6α-ethyl-7-keto-5β-cholan-24-oate (**9**), obtained as previously reported (500 mg, 1.2 mmol) [23], in a solution of MeOH: H_2_O 1:1 *v*/*v* (20 mL). The mixture was stirred for 4 h at reflux. The resulting solution was then acidified with HCl 6N and extracted with ethyl acetate (3 × 50 mL). The collected organic phases were washed with brine, dried over Na_2_SO_4_ anhydrous and evaporated under reduced pressure to give the carboxylic acid intermediate **10** (480 mg, quantitative yield).

Selected ^1^H-NMR (400 MHz, CD_3_OD): δ 1.25 (3H, s), 0.96 (3H, d, *J* = 6.4 Hz), 0.79 (3H, t, *J* = 7.2 Hz), 0.70 (3H, s). ^13^C-NMR (100 MHz, CDCl_3_): δ 215.1, 178.0, 54.7, 51.4, 50.7, 50.5, 48.7, 45.6, 43.2, 42.4, 37.5, 36.3, 35.1, 31.0 (2C), 30.3, 28.1, 26.7, 26.3, 24.9, 23.0, 21.3 (2C), 20.4, 18.8, 12.0. HR ESIMS *m*/*z* 403.3205 [M + H]^+^, C_26_H_43_O_3_ requires 403.3207.

#### 3.2.6. Synthesis of 6α-Ethyl-7α-hydroxy-24-nor-5β-chol-23-ene (**4**)

Compound **10** (250 mg, 0.62 mmol) was dissolved in dry toluene/dry pyridine (20 mL: 200 µL, 10:1 *v*/*v*) and Cu(OAc)_2_^.^H_2_O (372 mg, 1.8 mmol) was added in dark. After 30 min Pb(OAc)_4_ (1.3 g, 3.1 mmol) was added in dark. After 3 h the solution was heated to reflux for 1 h (no longer in the dark). The mixture was then cooled, and aqueous ethylene glycol was added. The resulting mixture was extracted with diethyl ether (3 × 50 mL). The combined organic phases were washed then with saturated solution of NaHCO_3_, water and brine. After drying over anhydrous Na_2_SO_4_, the residue was evaporated under vacuum to give 100 mg of product as crude residue. Then, the residue (100 mg, 0.28 mmol) was treated with a solution of dry THF (10 mL), at 0 °C dry methanol (76 µL, 1.9 mmol) and LiBH_4_ (980 µL, 4 M in THF, 1.9 mmol) were added. The resulting mixture was stirred for 2 h at 0 °C. The mixture was quenched by addition of 1M NaOH (1.24 mL) and then ethyl acetate. The organic phase was washed with water, dried (Na_2_SO_4_), and concentrated giving 91 mg of crude residue (91% yield). An analytic sample of compound **4** was obtained through purification with HPLC on a Synergi Fusion-RP C18 (4 µm; 4.6 mm i.d. × 250 mm), with MeOH/H_2_O (99:1) as eluent (flow rate 1 mL/min) (t_R_ = 8 min).

Selected ^1^H-NMR (400 MHz, CDCl_3_): δ 5.67 (1H, m), 4.91 (1H, d, *J* = 17.3 Hz), 4.82 (1H, d, *J* = 10.1 Hz), 3.70 (1H, br s), 1.04 (3H, d, *J* = 6.4 Hz), 0.90 (3H, s), 0.90 (3H, t ovl), 0.69 (3H, s). ^13^C-NMR (100 MHz, CDCl_3_): δ 145.2, 111.5, 71.2, 55.4, 50.6, 46.9, 42.7, 41.4, 41.3, 40.1, 39.6, 38.0, 36.3, 33.4, 28.5, 27.6, 24.3, 24.0, 23.8, 22.2, 21.3, 20.8, 20.1, 11.9, 11.7. HR ESIMS *m*/*z* 359.3305 [M + H]^+^, C_25_H_41_O requires 359.3308.

#### 3.2.7. Synthesis of 6α-Ethyl-7α-hydroxy-23,24-dinor-5β-cholane (**5**)

Compound **5** (35 mg, quantitative yield) was obtained using the compound **4** (35 mg, 0.1 mmol) as starting material in the same experimental conditions previously described for compound **2**. An analytic sample of compound **5** was obtained through purification with HPLC on a Synergy fusion-RP C18 (4 µm; 4.6 mm i.d. × 250 mm), with MeOH/H_2_O (99:1) as eluent (flow rate 1.5 mL/min) (t_R_ = 9.1 min).

Selected ^1^H-NMR (400 MHz, CDCl_3_): δ 3.71 (1H, br s), 0.92 (3H, d, *J* = 6.8 Hz), 0.90 (3H, s), 0.90 (3H, t ovl), 0.83 (3H, t, *J* = 7.3 Hz), 0.67 (3H, s). ^13^C-NMR (100 MHz, CDCl_3_): δ 71.2, 55.5, 50.6, 46.9, 42.6, 41.4, 40.1, 39.6, 37.9, 36.9, 36.4, 33.4, 28.2, 28.1, 27.6, 24.3, 24.0, 23.7, 22.2, 21.4, 20.8, 18.0, 11.8, 11.7, 10.3. HR ESIMS *m*/*z* 361.3461 [M + H]^+^, C_25_H_45_O requires 361.3465.

#### 3.2.8. Synthesis of 6α-Ethyl-7α-hydroxy-23,24-dinor-5β-cholan-22-ol (**6**)

Starting from the compound **4** (35 mg, 0.1 mmol), the compound **6** (34 mg, 95%) was obtained in the same experimental conditions previously described for compound **3**. An analytic sample of compound **6** was obtained through purification with HPLC on a Luna Omega Polar C18 (5 µm; 4.6 mm i.d. × 250 mm), with MeOH/H_2_O (92: 8) as eluent (flow rate 1.5 mL/min) (t_R_ = 19.3 min).

Selected ^1^H-NMR (400 MHz, CDCl_3_): δ 3.70 (1H, br s), 3.65 (1H, dd, *J* = 10.3, 3.0 Hz), 3.36 (1H, dd, *J* = 10.3, 7.0 Hz), 1.05 (3H, d, *J* = 6.7 Hz), 0.90 (3H, s), 0.90 (3H, t, *J* = 7.0 Hz), 0.69 (3H, s). ^13^C-NMR (100 MHz, CDCl_3_): δ 71.2, 68.0, 52.5, 50.3, 46.9, 42.8, 41.3, 40.5, 39.5, 38.7, 37.9, 36.3, 33.3, 27.7, 27.6, 24.3, 23.9, 23.8, 22.2, 21.3, 20.7, 16.7, 11.8, 11.7. HR ESIMS *m*/*z* 363.3255 [M + H]^+^, C_24_H_43_O_2_ requires 363.3258.

### 3.3. Biological Assays

#### 3.3.1. Cell Culture

HepG2, a human immortalized hepatocarcinoma cell line, was cultured and maintained at 37 °C and 5% CO_2_ in E-MEM additioned with 10% FBS, 1% glutamine and 1% penicillin/streptomycin. HEK293T and GLUTag cells, a murine intestinal endocrine cell line, were cultured and maintained at 37 °C and 5% CO_2_ in D-MEM additioned with 10% FBS, 1% glutamine and 1% penicillin/streptomycin.

#### 3.3.2. Transactivation Assay

To evaluate FXR mediated transactivation, HepG2 cells were transfected with 100 ng of human pSG5-FXR, 100 ng of human pSG5-RXR, 200 ng of the reporter vector p(hsp27)-TK-LUC containing the FXR response element IR1 cloned from the promoter of heat shock protein 27 (hsp27) and with 100 ng of pGL4.70 (Promega), a vector encoding the human Renilla gene. To evaluate GPBAR1 mediated transactivation, HEK293T cells were transfected with 200 ng of human pGL4.29 (Promega), a reporter vector containing a cAMP response element (CRE) that drives the transcription of the luciferase reporter gene luc2P, with 100 ng of pCMVSPORT6-human GPBAR1, and with 100 ng of pGL4.70. At 24 h post-transfection, cells were stimulated 18 h with CDCA, TLCA and compounds **1**–**6** (10 μM). In another experimental setting, at 24 h post-transfection, cells were stimulated with 50 μM of compounds in combination with CDCA or TLCA (10 μM). After treatments, 10 μL of cellular lysates were read using a Dual-Luciferase Reporter Assay System (Promega Italia srl, Milan, Italy) according manufacturer specifications using the Glomax20/20 luminometer (Promega Italia srl, Milan, Italy). Luciferase activities were assayed and normalized with Renilla activities.

#### 3.3.3. Dose–Response Curve on FXR and GPBAR1

To calculate the EC_50_ of FXR and GPBAR1, dose response curves were performed in HepG2 and HEK293T cells transfected as described above and then treated with increasing concentrations of compounds **1**, **2**, **3**, and **6** (from 0.5 to 50 µM). At 18 h post stimulations, cellular lysates were assayed for luciferase and Renilla activities using the Dual-Luciferase Reporter Assay System (E1980, Promega Italia srl, Milan, Italy). Luminescence was measured using Glomax 20/20 luminometer (Promega Iralia srl, Milan, Italy). Luciferase activities were normalized with Renilla activities.

#### 3.3.4. RNA Isolation and RT-PCR

HepG2 and GLUTag cells were plated at 1 × 10^6^ cells/well in a six well plate. After an overnight incubation, cells were starved and then stimulated for 18 h with compounds **1**, **2**, **3**, and **6**, 1 and 10 µM, or with CDCA or TLCA 10 µM. Total RNA was isolated from cells using the TRIzol reagent according to the manufacturer’s specifications (Invitrogen, Carlsbad, CA, USA). One microgram of purified RNA was treated with DNase-I and reverse transcribed with Superscript II (Invitrogen, Carlsbad, CA, USA). For Real Time PCR, 10 ng template was dissolved in 25 μL containing 200 nmol/L of each primer and 12.5 μL of 2× SYBR FAST Universal ready mix (Invitrogen, Carlsbad, CA, USA). All reactions were performed in triplicate, and the thermal cycling conditions were as follows: 2 min at 95 °C, followed by 40 cycles of 95 °C for 20 s and 60 °C for 30 s in StepOnePlus (Applied Biosystems, Foster City, CA, USA). The relative mRNA expression was calculated accordingly to the Ct method. Primers were designed using the software PRIMER31 using published data obtained from the NCBI database. Forward and reverse primer sequences were the following: human GAPDH, gaaggtgaaggtcggagt and catgggtggaatcatattggaa; human SHP, tctcttcttccgccctatca and aagggcttgctggacagtta; mouse GAPDH, ctgagtatgtcgtggagtctac and gttggtggtgcaggatgcattg; mouse pro-glucagon, tgaagacaaacgccactcac and caatgttgttccggttcctc.

### 3.4. Physiochemical Properties and Pharmacokinetic Characterization

#### 3.4.1. LC-MS/MS ADME Methods

Chromatography was performed using an Alliance pump system coupled to a Q-ToF Premiere (Waters Co., Milford, MA, USA). The mixture was separated on a Luna 5 µm C8(2) 100° A 150 × 2 mm from Phenomenex. The mobile phase consisted of 0.2% formic acid (FA) in water as solvent A and 0.2% FA in acetonitrile as solvent B at a flow rate of 200 μL/min. The gradient was as follows: 0–2 min (70% A and 30% B), 2–20 min (5% A and 95% B), 20–30 min (70% A and 30% B). The detection of analytes was achieved by electrospray ionization (ESI) in the positive mode with the appropriate MS/MS transitions, if necessary.

#### 3.4.2. Solubility Measurements

Ten microliters of a 10 mM solution in DMSO of each compound was diluted either in 490 μL of PBS pH 7.4 or in organic solvent MeOH (in triplicate). The tubes were gently shaken 24 h at room temperature, then centrifuged for 5 min at 4000 rpm. 10 microliters of sample were diluted in 490 μL of MeOH. The solubility is determined by the ratio of mass signal area PBS/organic solvent.

#### 3.4.3. Microsomal Stability

Male mouse (CD-1) liver microsomes (Sigma-Aldrich, St. Louis, MO, USA) were used. All incubations were performed in duplicate in a shaking water bath at 37 °C. The incubation mixtures contained 1 μM compound with 1% DMSO used as a vehicle, mouse liver microsomes (0.3 mg of microsomal protein per mL), 5 mM MgCl_2_, 1 mM NADP, 5 mM glucose 6-phosphate, 0.4 U·mL^−1^ glucose 6-phosphate dehydrogenase, and 50 mM potassium phosphate buffer (pH 7.4) in a final volume of 0.5 mL. Aliquots were removed at 0, 5, 10, 20, 30, and 40 min after microsome addition and the reaction was stopped by adding 200 μL of ice-cold acetonitrile [28]. After two hours, the samples were centrifuged for 10 min at 10,000 rpm, and the supernatants were transferred in matrix tubes for LC-MS/MS analysis. LC-MS/MS analysis were carried out setting the *m*/*z* window around the value of the MH^+^ of the unmodified compounds.

Propranolol, known as a high hepatic clearance drug in rodents, was used as a quality-control compound for the microsomal incubations. The slope of the linear regression of the curve obtained reporting the natural logarithm of compound area versus incubation time (−k) was used in the conversion to in vitro t_1/2_ values by t_1/2_ = −ln(2)/k. In vitro intrinsic clearance (Cl_int_ expressed as μL/min/mg) was calculated according to the following formula: Cl_int_ = volume of reaction (µL)/t_1/2_(min)/protein of liver microsomes (mg). The percentage of unmodified compound has been calculated assuming the area of the compound peak at time 0 min as 100%.

### 3.5. Molecular Docking

The Glide [36] software was used to perform molecular docking calculations in the crystal structure of the *Rattus Norvegicus* FXR-LDB (PDB code 1osv) [32] and in the homology model of human GPBAR1, previously developed by us [17]. We note that the *Rattus Norvegicus* FXR-LBD shares the 95% of homology with the human FXR-LBD, with all of the residues in the ligand binding site conserved among the two species. Therefore, the *Rattus Norvegicus* FXR-LDB X-ray structure can be profitably used to study the binding mode of ligands of human FXR. Protein and ligand structures were prepared as described in previous papers [33,37]. For each receptor, a box of 30 Å × 30 Å × 30 Å centered on the ligand binding cavity was initially created to compute the interaction grids. Upon docking calculations, ligands macrocyclic rings were treated as rigid; otherwise, default parameters were applied. The standard precision (SP) mode of the GlideScore function was used to score and rank the predicted binding poses [38,39]. For each ligand, the best 10 docking poses were considered for visual inspection. For FXR, all the residue numbers were taken from the wild-type sequence of FXR.

## 4. Conclusions

In conclusion, in this report, a new series of nonacidic 6-ethylcholane derivatives has been designed and synthesized and their in vitro activities on FXR and GPBAR1 were assayed. This study resulted in the identification of compound **6**, a potent and selective FXR agonist with improved metabolic stability in vitro, and of several derivatives showing potent dual agonistic activity. Of interest, the lack of acidic side chains, that can help to preserve their specificity over several off-targets, as well as their favorable ADME profiles make compounds **1**–**3** and **6** suitable for further development.

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
