# Peer review of "Introduction of Nonacidic Side Chains on 6-Ethylcholane Scaffolds in the Identification of Potent Bile Acid Receptor Agonists with Improved Pharmacokinetic Properties"

_molecules, 2019, doi:10.3390/molecules24061043_

Reviewer 1 Report

In this study, the authors modified the existing side chain scaffold of a dual agonist at bile receptors (6-ECDCA) and characterized the pharmacology and pharmacokinetic of these newly synthesized compounds. The authors found that compounds 1-3 displayed agonistic activity at both FXR and GPBAR1 receptors, with improved pharmacokinetic compared to reference compound, 6-ECDCA. In addition compound 6 showed selective activity at the FXR receptor, again with improved pharmacokinetic compared to 6-ECDCA.

Overall, the manuscript is well written and clear in background/results. However, the authors should address the following comments:

- the authors should state that the compounds have been tested in different receptor orthologs for both FXR and GPBAR1. For example, results in table 1 were generated by transfecting the human orthologs of the two receptors, while figure 2 was generated using murine cells physiologically expressing the murine orthologs of the same receptors. This should be clearly stated in the text. In addition, this might help explaining the reason why the efficacies of compound 1-3 are much higher compared to reference compound CDCA in table 1, but not in Figure 3.

-The authors should also state what receptor orthologs were used for the docking studies? Human or mouse?

Minor concerns.

-it is not clear in the introduction whether 6-ECDCA is a dual agonist at the bile receptors. A sentence should be added.

-the EC50 of the reference compounds, CDCA and TLCA, should be reported in order to compare it with the newly synthesized compounds. Also, in the same set of experiments the authors should also test the dual agonists, 6-ECDCA as a second reference compound.

- Standard deviations in table 1 should be added.

- Figure 2 should report also the effect of compound 6.

- mRNA expression is reported as relative to NT condition. If this is the case than the NT condition should be equal to 1.

- Some of the abbreviations used should be defined (e.g. OCA, PBC, Clint)

- In line 135, the authors should change “potent” with “efficacious”. Same for line 84, where the word “effective” should be change with “efficacious”

Author Response

-The authors should state that the compounds have been tested in different receptor orthologs for both FXR and GPBAR1. For example, results in table 1 were generated by transfecting the human orthologs of the two receptors, while figure 2 was generated using murine cells physiologically expressing the murine orthologs of the same receptors. This should be clearly stated in the text. In addition, this might help explaining the reason why the efficacies of compound 1-3 are much higher compared to reference compound CDCA in table 1, but not in Figure 3.

Answer.

We agree with referee and accordingly we added her-his suggestions on pages 3-5 and 9-10 on the revised version.

-The authors should also state what receptor orthologs were used for the docking studies? Human or mouse?

Answer.

We agree with the Reviewer and we have added to the docking paragraph in the Methods section the following information:

“The Glide [36] software was used to perform molecular docking calculations in the crystal structure of the Rattus Norvegicus FXR-LDB (PDB code 1osv) [32] and in the homology model of human GPBAR1, previously developed by us [17]. We note that the Rattus Norvegicus FXR-LBD shares the 95% of homology with the human FXR-LBD, with all of the residues in the ligand binding site conserved among the two species. Therefore, the Rattus Norvegicus FXR-LDB X-ray structure can be profitably used to study the binding mode of ligands of human FXR.”

-it is not clear in the introduction whether 6-ECDCA is a dual agonist at the bile receptors. A sentence should be added.

Answer.

Please see lines 54-55 in the revised version with also the EC50 for 6-ECDCA added.

-the EC50 of the reference compounds, CDCA and TLCA, should be reported in order to compare it with the newly synthesized compounds. Also, in the same set of experiments the authors should also test the dual agonists, 6-ECDCA as a second reference compound.

Answer.

We have added EC50 of CDCA, TLCA and 6-ECDCA in the text and in the Table 1. Moreover, we have also tested in the same experiments the 6-ECDCA and we reported the results in Table 1.

-Standard deviations in table 1 should be added.

Answer.

Done

-Figure 2 should report also the effect of compound 6.

Answer.

We didn’t include compound 6 in the RT-PCR because it was no-active in transactivation assay, as reported in Table 1.

-mRNA expression is reported as relative to NT condition. If this is the case than the NT condition should be equal to 1.

Answer.

As requested, we have changed figure 2 and figure 3.

- Some of the abbreviations used should be defined (e.g. OCA, PBC, Clint)

Answer.

We have added the definition of the abbreviations.

- In line 135, the authors should change “potent” with “efficacious”. Same for line 84, where the word “effective” should be change with “efficacious”

Answer.

Done

Reviewer 2 Report

In this manuscript, the authors report novel synthetic FXR agonists. Structural-activity relationships for FXR and GPBAR1 agonistic activities were studied well. It's clearly shown that compounds 1-3 are dual agonists and that compound 6 is a FXR agonist. I have a minor comment.

1. Please discuss how compound 6 is superior to the existing FXR agonists, such as 6-ECDCA (INT-747). How about compounds 1-3 if compared to the reported dual agonist, such as INT-767? Please explain the advantage of novel compounds over the previously reported ones.

Author Response

-Please discuss how compound 6 is superior to the existing FXR agonists, such as 6-ECDCA (INT-747). How about compounds 1-3 if compared to the reported dual agonist, such as INT-767? Please explain the advantage of novel compounds over the previously reported ones.

Answer.

We acknowledge the reviewer. Our molecules (compounds 1-3 and 6) are similar to 6-ECDCA in the chemical structure and in the binding mode at FXR, even if less efficacious and potent respect to 6-ECDCA. Indeed the target of our work was the identification of derivative with improved pharmacokinetics. As showed, the modification on the side chain as well as on the ring A of steroidal scaffold produces promising improvement in term of metabolic stability, without any change in aqueous solubility.

Reviewer 3 Report

This paper presented the synthesis of six nonacidic 6-ethyl cholane derivatives designed to expand the chemical diversity of bile acid receptor modulators. The work is interesting, informative and well-written, and I really enjoyed reading it. In my opinion, it deserves to be published without any serious modifications. I only have a minor suggestion of adding a list of abbreviations.  

Author Response

I only have a minor suggestion of adding a list of abbreviations.  

Answer.

As the word template requires, the abbreviations should be defined in parentheses the first time they appear in the abstract, main text, and in figure or table captions. We have been careful to introduce all omitted abbreviations.

Reviewer 4 Report

The paper entitled “Introduction of nonacidic side chains on 6-ethylcholane scaffolds in the identification of potent bile acid receptor agonists with improved pharmacokinetic properties” by Finamore et al., presents the synthesis of new nonacidic 6-ethyl cholane derivatives and their in vitro activities on FXR and GPBAR1. The authors described new interesting molecule 6, a potent and selective FXR agonist with improved metabolic stability in vitro with dual agonistic activity. Moreover, the favorable ADME profiles of compounds 1-3 and 6 were also confirmed. Thus, the work deserves to be published in Molecules after the following minor revision:

1.      Terms must be defined each time they are first used in the text. For instance, several names of compound 6-ECDCA (obeticholic acid, Ocaliva) were shown whereas “OCA”, which appears in next sentence was not included. Please define also the following term: UDCA,  NASH, PBC, 7-KLCA etc.

2.      Some editorial errors were found.  Please unify following differences:

P1 line 34    (Figure 1)

P1 line 41    (Fig. 1)

P2 line 51    (Fig 1)

P3 line 81    (Table 1)

P3 line 89    (table 2)

Another erros found:

P3 line 113  “Sulfatation” should be “Sulfation”

P3 line 342   “epatocarcinoma” should be "hepatocarcinoma”

P10 line 368     “ 1 × 106 cells/well”-  “6” should be in superscript

3.      Please, include “Scheme 1” into the text to clarify where this scheme is described

4.      Figures and schemes quality should be improved

5.      The “Metabolic stability” chapter should be improved. Please include in the discussion the Clint or t½ values. Some information about the obtained metabolites of 1-3 and 6 should be provided and compared to the ECDCA ( number of metabolites, molecular masses). Did the sulfation at C3  occurred for ECDCA after incubation with rat liver microsomes?

Author Response

-Terms must be defined each time they are first used in the text. For instance, several names of compound 6-ECDCA (obeticholic acid, Ocaliva) were shown whereas “OCA”, which appears in next sentence was not included. Please define also the following term: UDCA, NASH, PBC, 7-KLCA etc.

Answer.

Done

-Some editorial errors were found.  Please unify following differences:

-Please, include “Scheme 1” into the text to clarify where this scheme is described

-Figures and schemes quality should be improved

Answer.

As requested, the manuscript has been carefully revised and all original figures have been replaced with production quality figures.

-The“Metabolic stability” chapter should be improved. Please include in the discussion the Clint or t½values.

Answer.

In lines 124-129 the Clint and t1/2 are now reported for compounds 3 and 6, together with their definition.

-Some information about the obtained metabolites of 1-3 and 6 should be provided and compared to the ECDCA (number of metabolites, molecular masses). Did the sulfation at C3 occurred for ECDCA after incubation with rat liver microsomes?

Answer.

We measured the microsomal stability of our compounds in male mouse (CD-1) liver microsomes. In the above assay, to be more sensible, LCMSMS analysis was carried out setting the m/z window around the value of the MH+ of the unmodified compounds (as now reported in M&M). Indeed the concentration of each metabolites is very low in accordance with the protocol reported in literature [28] and we measured principally the compound reduction. However, as demonstrated by Alnouti [27], two pathways are involved in bile acid metabolism and in structurally-correlated small molecules such as 6-ECDCA: 1) sulfation at C-3 alfa OH and 2) amidation at the COOH on the side chain with glycine in human and taurine in mouse. Thus, since our compounds are differently modified at those positions, it is absolutely plausible that their impressive improvement in term of metabolic stability is due to the lack of this/these point/s of metabolization.